# IoT-Based SHM Using Digital Twins for Interoperable and Scalable Decentralized Smart Sensing Systems

Jiahang Chen [1,*] , Jan Reitz [1] , Rebecca Richstein [2] , Kai-Uwe Schröder [2] and Jürgen Roßmann [1]

1 Institute for Man-Machine Interaction, RWTH Aachen University, Ahornstr. 55, 52074 Aachen, Germany; reitz@mmi.rwth-aachen.de (J.R.); rossmann@mmi.rwth-aachen.de (J.R.)
2 Institute of Structural Mechanics and Lightweight Design, RWTH Aachen University, Wüllnerstraße 7, 52062 Aachen, Germany; rebecca.richstein@sla.rwth-aachen.de (R.R); kai-uwe.schroeder@sla.rwth-aachen.de (K.-U.S.)
* Correspondence: chen@mmi.rwth-aachen.de

**Abstract:** Advancing digitalization is reaching the realm of lightweight construction and structural–mechanical components. Through the synergistic combination of distributed sensors and intelligent evaluation algorithms, traditional structures evolve into smart sensing systems. In this context, Structural Health Monitoring (SHM) plays a key role in managing potential risks to human safety and environmental integrity due to structural failures by providing analysis, localization, and records of the structure's loading and damaging conditions. The establishment of networks between sensors and data-processing units via Internet of Things (IoT) technologies is an elementary prerequisite for the integration of SHM into smart sensing systems. However, this integrating of SHM faces significant restrictions due to scalability challenges of smart sensing systems and IoT-specific issues, including communication security and interoperability. To address the issue, this paper presents a comprehensive methodological framework aimed at facilitating the scalable integration of objects ranging from components via systems to clusters into SHM systems. Furthermore, we detail a prototypical implementation of the conceptually developed framework, demonstrating a structural component and its corresponding Digital Twin. Here, real-time capable deformation and strain-based monitoring of the structure are achieved, showcasing the practical applicability of the proposed framework.

**Keywords:** IoT; smart sensing systems; structural health monitoring; digital twins





## 1. Introduction

The primary function of lightweight structural components is to provide rigidity and strength while minimizing weight. Recent approaches have furthered their functionality by integrating distributed sensors and intelligent algorithms, evolving into smart sensing systems. These systems transform previously passive structural components into active components so that theycan digitally process data and exchange their state, for example, operational status, with each other. Presently, smart sensing systems are applied in various domains like water quality monitoring [1], generalized environmental monitoring [2], healthcare monitoring [3], and human motion disorders [4], showing their innovation in data collection and analysis. Back to the realm of lightweight structural components, typical structures that can be retrofitted to smart sensing systems include bridges [5], aircraft [6], or wind turbines [7]. Such integrated systems, comprised of structural components, sensors, and data-processing systems, can potentially pose risks to human safety and environmental integrity due to system failures [8]. In response, Structural Health Monitoring (SHM) has emerged with the principal aim of analysis, localization, and recording of the loading and damaging conditions, enabling the prediction of systems' remaining useful life [9].

The implementation of SHM necessitates the establishment of networks that connect sensors, digital models, evaluation algorithms, and users. Here, data acquisition, data

processing, and data exchange are essential concerns. Traditional data acquisition systems use wires to connect sensors to a centralized server (e.g., database), where data processing and interpreting are carried out. Newer applications advocate so-called wireless sensing networks (WSN) to reduce costs related to installation and maintenance [10]. Overall, technological advancements and the widespread availability of wireless networks have shifted SHM from wire-based methods to real-time WSNs using wireless communication protocols [11]. Beyond WSNs, the Internet of Things (IoT) technologies are not limited to localized data processing and energy-efficient communication. They leverage a more complex and hierarchical architecture, integrating various devices, cloud-based services, and users to enable a seamless information flow across the global network. The integration of IoT paradigms drives more innovative solutions of communication in the field of SHM, enabling remote and continuous monitoring, as reflected in recent research [12–15].

Nevertheless, based on the observation of the current IoT-based SHM systems, we can generally identify the following issues. Using IoT technologies comes with significant challenges regarding communication security (e.g., related to confidentiality, integrity, and availability), data sovereignty (i.e., related to the control over the data), and exchange interoperability (e.g., how can heterogeneous data and services be understood at the same level?). These considerations are often only an afterthought, but they are crucial to apply SHM in practice. Moreover, structures of smart sensing systems are becoming increasingly complex, requiring multiple measurement points or sensors even when monitoring hot-spot regions. Here, we identify the absence of a general approach to enable the monitoring and management of individual components and scaling to systems or clusters in the IoT.

The center of the proposed scheme is Digital Twins (DTs)—virtual representations of physical assets. Until now, the concept of DTs is loosely defined, as there are diverse standards to interpret how a DT should look like and which functions it should have. Precisely because of this loose definition, DTs are highly expandable. This leads to the fact that the development of DTs is closely related to specific needs. In several current DT paradigms, DTs are associated with different security threats. Literature like [16,17] intents to classify these threats and shows the DT's potential to ensure appropriate and trustworthy data exchange in a secured way. Meanwhile, DTs have gained traction in the digital transformation of various domains. This term is also relevant to structural engineering, as digitalization transforms how structures are designed, managed, and maintained. SHM applications, ranging from estimating component lifetime to maintenance scheduling, benefit from the incorporation of DTs, enhancing interoperability across diverse scales due to consistent and uniform modeling of DT's structure and interface [18]. Studies focusing on the use DTs in SHM systems can be classified based on their application focus, like data analysis of the measured data [19–21], communication efficiency [22–24], and formulation of the integrated process [25] or the formalized modeling of DTs [26]. We find few publications that use DTs to address integration scalability in SHM.

Based on these observations, we propose an organizational scheme to describe structural–mechanical objects from individual components to clusters based on hierarchical DTs. Additionally, we present the conceptual framework for integrating DTs into IoT-based SHM systems, addressing data security and interoperability. The remaining parts of the paper are structured as follows: Section 2 reviews published SHM applications with a focus on the employed networking infrastructures, as well as existing organizational schemes in SHM systems. In Section 3, we propose an organizational scheme of IoT-based SHM systems and the respective conceptual framework. Finally, a proof-of-concept implementation will be demonstrated in Section 4 to monitor the operational of an exemplary cantilever via an app in near real time. Section 5 discusses the extension possibilities of the prototypical application using our methodological approach. Section 6 concludes this paper.

## 2. Related Work

This section provides an overview of pivotal publications in the domains of Structural Health Monitoring (SHM) and the IoT. First, we focus on general, higher-level organi-

zational schemes that structurally describe SHM systems, considering their IoT-based interconnections. The overall objective is to extract the SHM's central requirements for interconnections through IoT infrastructures. Based on that, we review and compare IoT infrastructures concerning their suitability for SHM.

### 2.1. Organizational Schemes of SHM Systems

A review of current publications reveals that IoT-based SHM systems tend to adopt layered architectures. Aguzzi et al. [27] subdivide their Web of Things platform for SHM application into four distinct layers: A monitoring layer directly connected to physical structures, an edge layer focusing on data acquisition and preprocessing, a data management layer addressing storage, aggregation, and visualization of acquired data, and a data analytics layer dedicated to interpreting the data for condition assessment and damage detection, localization, and prediction. Similar to this layered approach, Lamonaca et al. [28] present their SHM system as an aggregation of interconnected smart objects. They define a dual-layered architecture: a physical layer encompassing all sensors and actuators that comprise smart objects and a cyber part responsible for data processing and communication. The cyber part is further subdivided into functional layers targeting signal processing, event detection, and real-time applications. Similar layered architectures can also be found in other publications, e.g., [29,30].

The aforementioned organizational proposals focus on subdividing the data stream linearly from data acquisition via (central) evaluation to end-user visualization. The modeling granularity of physical objects remains unrefined. Physical objects—in this case, structures—can be increasingly complex to cover use cases ranging from basic elements to a complicated wind turbine. Hence, they demand a more detailed approach to represent their hierarchical and structural complexity accurately. Additionally, current proposals predominantly emphasize vertical communication along the data stream, i.e., traversing from the physical object (data acquisition) through a cloud-based analysis service (data preprocessing and aggregation) to the end user (data visualization). However, there is an oversight in horizontal communication—the interconnection of objects, regardless of the complexity. Enabling horizontal interconnections between objects is crucial, especially in complex SHM systems consisting of different structures and sensors. Here, individual objects should be able to autonomously manage their data and facilitate interfaces for interconnections. The use of IoT paradigms can help in creating organizational schemes for SHM applications in this respect.

### 2.2. The Role of DTs

Aspects and concepts that shape the definition of DTs are diverse. In general, the summary of characteristics published by Jones et al. [31] is widely accepted. We perceive DTs as virtual representations of physical entities and are realized by aggregation of computation and communication technologies. In the digital world, DTs utilize their metadata to describe the basic physical structure, operational state, provided service functions, as well as access properties and interfaces. They are assigned a globally unique identity and equipped with communication endpoints so that peer-to-peer communication is possible. When it comes to IoT, we consider both the interconnection of DTs themselves as well as the interconnection of DTs with other IoT objects and end users. Hence, DTs are also regarded as communication nodes with a unique addressable and available identity in this decentralized landscape. In this context, the focus of DTs shifts to interconnection and holistic modeling [32], rather than other popular aspects like 3D modeling, product life cycle management, and so on. Interconnection seeks a seamless connection to collaborate on shared targets, which requires uniform interfaces and an identical understanding of the communication protocols used. Holistic modeling intends to formally and semantically describe DTs' physical structure, functional composition, and other features using a standardized data model [33]. In this context, a specific example is the integration of a structure

into a building information model [21]. In the domain of structure mechanics, a general framework for implementing DTs has not been established yet, as stated in [32].

### 2.3. Dimensions of IoT-Based SHM Systems

#### 2.3.1. Interoperability

Interoperability in the IoT, as defined by Konduru et al. [34], refers to the ability of diverse connected objects to communicate at the same technical and semantic level. It is a crucial foundation for scalable and flexible IoT systems, enabling the integration of new objects and technologies without disrupting existing communication paradigms. Achieving interoperability requires semantics in both object structures and communication languages.

In the context of SHM, Aguzzi et al. [27] utilize the Thing Description format of the W3C's WoT specification to structurally describe SHM systems. Based on Thing Description, a semantic layer can be directly added to the meta model to realize interoperable data exchange. Similar ideas can be found in the publication from Gigli et al. [35], which proposes the semantic formalization of exchanged data.

#### 2.3.2. Offline Capability

The capability to continuously provide intended functions (e.g., data processing) is pivotal for long-term SHM systems [9]. This capability should be kept even in offline scenarios [30]. However, offline capability in wireless SHM systems presents challenges since it necessitates wireless transmission of entire structural response data sets, which has been proven to negatively impact the autonomy of wireless sensor nodes [36]. Investigating how IoT technologies and DTs are combined in SHM systems illustrates the feasibility of maintaining offline capabilities even in offline scenarios or disrupted network connections [30,37].

#### 2.3.3. Decentralized Data Collection and Centralized Data Analysis

SHM processes, as interpreted by Farrar et al. [9], require a dynamic approach to data acquisition. Due to the inherent variability in structural geometries, damage manifestation can occur either locally within specific areas or more broadly across spatially distributed locations. Decentralized data acquisition is essential to address this variability and allow for precise monitoring of structural conditions. Moreover, it may be necessary to perform some data preprocessing on-site (e.g., through edge computing) to ensure that the data being collected is of high quality [38].

Centralized data analysis is required to consolidate data collected from decentralized sources, especially when diverse sensor types from different vendors are being used. Integrating data into a unified analytical framework facilitates a comprehensive view of the structure's condition and enhances the ability to identify issues [38].

Both decentralized data collection and centralized analysis underscore the need for an IoT infrastructure that supports seamless connectivity across various entities, such as smart sensors or cloud-based services. The combination of these two paradigms within an IoT infrastructure not only enhances the accuracy and efficiency of SHM systems but also ensures that data acquisition and processing are scalable and adaptable to the evolving structural conditions.

#### 2.3.4. Flexibility and Scalability

As stated in Section 2.3.3, SHM systems frequently employ various sensors to monitor different aspects of structures. As structures and their monitoring needs may change over time [9], these systems should be capable of expanding or contracting by adding or removing sensors, visualization units, and cloud-based services with ease. Additionally, SHM systems must be established with the flexibility to integrate new technologies, including novel communication standards and sensors, as they are introduced [39].

### 2.3.5. Secure Communication

In the IoT, open sharing of information is crucial for enhancing collaboration between different systems.

However, this openness requires trust, which can be established by strong security properties of the underlying IoT infrastructure. Communication security is a crucial concern in the field of IoT-based SHM [15], which can typically be evaluated using the confidentiality, integrity, and availability (CIA) triad [40].

In general, a robust security concept that satisfies CIA depends on appropriate and comprehensive authentication and authorization methods, such as OpenID Connect [41] and access control [40]. They help systems to ensure that data are protected from unauthenticated and unauthorized access. To further strengthen the framework, an in-depth exploration of data privacy approaches is necessary. Privacy-preserving techniques, such as differential privacy and anonymization of sensor data, are dedicated to safeguarding user data against misuse. This is particularly significant for machine manufacturers, who would not allow measurement data from their potentially "damaged" components to be publicly available. Meanwhile, it ensures that SHM systems associated with IoT technologies are secure and resistant to manipulation or tampering.

### 2.4. Existing IoT Infrastructures for SHM Systems

In light of the dimensions illustrated in Section 2.3, we select and analyze both open-source and commercial IoT infrastructures (for more, see [42]) which have been (or can be) used in IoT-based SHM systems, see Table 1 for a comparative overview.

Bosch IoT Things [43] provides a versatile IoT infrastructure, facilitating the management of DTs for their IoT devices (assets). This infrastructure allows DTs to be equipped with various communication interfaces, enabling a bidirectional connection with their physical counterparts to manage asset data, obtain notifications on all relevant changes, and keep in a synchronous state with their assets. However, the modeling of DTs and the message utilized for communication is not the focus of Bosch IoT Things. This leads to interoperability issues when connected smart sensors are from different types or vendors. Although data acquisition is conducted in a decentralized manner, DTs and their measurements are forced to reside in the centralized cloud. However, data aggregation is still not centralized but dispersed within each DT aggregated in the same cloud. Due to the partial open-source nature, expanding the number of connected DTs involves the associated cost, consistently hindering flexibility and scalability. Security is considered from the transport layer to application-level access control, comprising device authentication via X.509 and application access control via OpenID Connect.

Commercial IoT infrastructures like Siemens Mindsphere [44] and Microsoft Azure [45] predominately offer centralized cloud-based databases for data storage and management [46]. These infrastructures cater to a wide range of IoT applications, such as machine learning. However, their primary focus is placed on enhancing cross-company or cross-sector value chains. For example, the primary area of Siemens Mindsphere is providing functionalities and technologies for digital services in industrial manufacturing controlled by Siemens PLC. Hence, there remains the question of the general applicability of those commercial IoT infrastructures in SHM systems. In the context of interoperability, there are defined data models for DTs. The Digital Twin Definition Language (DTDL) [47] from Microsoft Azure provides a concept to model DTs with self-defined vocabulary and aspects. This language makes the hierarchical modeling of the SHM system more flexible. Towards security, there are implementations observed in different layers by those infrastructures [46].

ThingsBoard IoT is an open-source IoT infrastructure consisting of infrastructure components, databases, and gateways. This infrastructure can enable the out-of-the-box IoT cloud (i.e., plug-and-play cloud-based applications) or on-premises solution with different communication protocols [48]. It implies a flexible integration of, e.g., a cloud-based centralized data aggregation to gain an insight into heterogeneous data [49]. As for data acquisition, Ismail et al. [50] show the performance on the throughput of the

platform, evaluated with REST and MQTT. Both results prove the data collection capability in decentralized scenarios. Moreover, this infrastructure allows users to add individual functionality as well as rules for diverse workflows. In the context of DTs' integration, the system interprets the general lack of communication interoperability since it is necessary to define a conceptual data model for devices and communication protocol to interpret the meaning of exchanged messages. This aims to make the communication between devices understandable at the same technical and semantic level. From the perspective of IoT security, the IoT platform provides the options associated with device authorization flow by access token, x.509, and MQTT basic credentials.

The Smart Systems Service Infrastructure (S3I, see [51,52]) is an open-source IoT infrastructure with a few centralized software services. This infrastructure allows the connected objects to authenticate and authorize themselves via S3I IdentityProvider, store and refind their meta information, including properties and service functions via S3I Directory, and communicate end-to-end compliantly with each other via S3I Broker. Basically, the infrastructure is originally dedicated to forestry applications, enabling decentralized interconnection between Forestry 4.0 things (F4.0 things), consisting of physical assets and their DTs, software services, and apps. However, S3I's distribution-oriented design (i.e., retaining as little central architecture as possible, allowing communication logic to be decentralized for execution) makes it possible to use S3I in other domains as well. In the S3I, things are not enforced to transmit and store their data to the centralized infrastructure; they only send metadata information to the S3I Directory so that things can be searched and discovered. As proposed by Chen et al. [53], the use of the S3I ensures security during communication since a comprehensive method is provided towards confidentiality, integrity, and availability, from OAuth 2.0-based authentication, authorization up to end-to-end encrypted communication via the S3I Broker. Moreover, the control over data is always kept since decentralized connected things only expose an interface to the outside, protected with fine-grained access control. Towards interoperability, the conceptual meta data model of the S3I Directory is delivered to allow the structure and content of F4.0 things to be mapped into the meta model. The S3I-B protocol specifies the predefined structure of S3I-B messages (including user messages, service messages, attribute messages, etc.) exchanged between different F4.0 Things [52]. Together with the forest modeling language 4.0 (ForestML 4.0, see [54]), decentralized F4.0 things (thus, also DTs) are described structurally and formally. Hence, technical interoperability during communication can be ensured.

**Table 1.** Comparison of IoT Infrastructures used for SHM systems.

| Features | Bosch IoT Things [55] | Microsoft Azure [56] | ThingBoard [57–59] | S3I [51–53] |
|---|---|---|---|---|
| Interoperability | - | + | o | ++ |
| Offline Capability | + | ++ | + | ++ |
| Decentralized Data Collection | ++ | ++ | ++ | ++ |
| Centralized Data Aggregation | o | ++ | ++ | o |
| Flexibility and Scalability | - | - | + | ++ |
| Communication and Data Security | + | + | + | + |
| Open Source | o | - - | ++ | ++ |

## 3. Concept

Drawing upon the literature review in Section 2, we propose a decentralized SHM system methodology with the term DT as the central abstraction. DTs serve as a bridge connecting various stakeholders, services, and data, regardless of their technical implementations or locations, whether in edge devices, cloud environments, or a hybrid of both. This interconnectedness is crucial as it transforms isolated technical components into a cohesive, value-added network.

The proposed system establishes a one-to-one relationship between DTs and physical objects, ranging from individual components to clusters. This granularity in representation not only refines the system's monitoring capabilities but also reduces implementation redundancy, therefore enhancing the flexibility of the SHM system (Section 3.1).

Interoperability is vital for the system's functionality, necessitating the capacity of different DTs to coordinate with each other seamlessly. This cooperation is largely dependent on the DTs' ability to understand each other, facilitated by standardized description structures and a common interaction language (Section 3.2).

Each DT is designed to support specific functionalities such as data preprocessing, storage, and analysis. The system's unified interface and interoperability protocols ensure that these DTs can be integrated to address more complex monitoring tasks, meeting a diverse range of SHM criteria (Section 3.3).

### 3.1. Hierarchical Structure of the Proposed SHM System

The proposed IoT-based SHM system is organized hierarchically and structured into layers of increasing abstraction. Each layer provides unique functionalities and insights. As illustrated through the example of wind power (see Figure 1), this system starts at the component level, including rotor blades, towers, and nacelles. These elements form wind turbines, which are then grouped into wind parks.

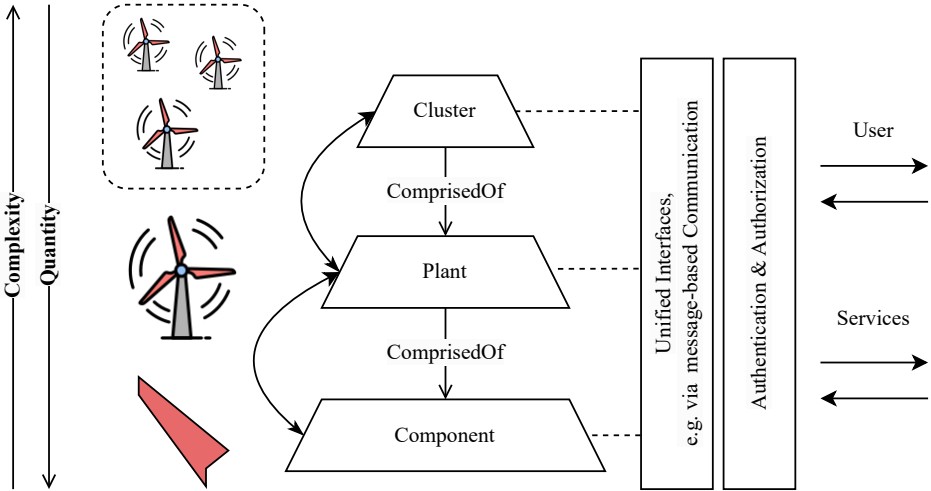

**Figure 1.** Vision from physical objects: scales in IoT networks for SHM systems.

### 3.2. Data Model for IoT-Based SHM Systems

The proposed system is comprised of a component level, a plant level, and a cluster level. At the component level, the system focuses on acquiring, processing, and interpreting sensor data. This level addresses specific questions, such as determining the next maintenance time for the nacelle or detecting deformations in blades exceeding certain thresholds. The plant level aggregates data from its components, offering a consolidated view of overall health and service requirements. Furthermore, the clustering of plants, e.g., clustering wind turbines into a wind park, represents an organizational level where data are aggregated from multiple plants. As we move up the hierarchy, the number of objects decreases, but the complexity increases. Although the number of clusters is smaller,

they contain numerous systems, which in turn are made up of many components. Despite this complexity at higher levels, it remains manageable at the component level due to the separation of concerns into individual DTs.

Thus, the component level is critical for direct interaction with sensors and raw structural data, forming the foundation of the SHM system. Higher levels, such as the plant or cluster levels, leverage these data, offering broader insights and facilitating interactions between components, plants, and the environment. The primary role of the cluster level is data consolidation, while models and detailed investigations are most effectively carried out at the plant level. This hierarchical structure ensures efficient data management and analysis across different levels of SHM systems.

In decentralized IoT-based SHM systems, the necessity for a shared data model is paramount, especially to ensure both technical and semantic interoperability among diverse system components. A shared data model facilitates consistent communication, data exchange, and understanding across various elements within the system, regardless of their designs or functions. This is particularly crucial in systems where components from different vendors or with varying technical specifications must work cohesively.

Our proposed data model is an augmentation of the established ML 4.0 data model [54], tailored specifically for the structural–mechanical domain. This data model is used to provide a comprehensive overview regarding the hierarchical physical structures, digital functionality, interfaces, and associated properties, as well as aggregates this information into a DT. We term this extension the mechanical modeling language 4.0 (mml40). Figure 2 illustrates this extension, highlighted in red, in juxtaposition with the original elements of the ML 4.0 model, depicted in gray. The central element is the (*ml40::Thing*), which represent independent units in the IoT. Each thing encompasses roles (*ml40::Role*) and features (*ml40::Features*), enabling a detailed and functional characterization, e.g., properties and services of each unit. In the mml40 extension, things in the field of SHM are further specialized into components (*mml40::Component*), plants (*mml40::plant*), and clusters (*mml40::Cluster*), therefore aligning the data model with the hierarchical structure of the proposed SHM system in Section 3.1. To add a dynamic aspect to this model, we introduce the concept of events to the data model. This allows the modeling of events to be autonomously published by things alongside the publishing conditions, event content, and meaning. Key features of these events include the topic (serving as the event identifier), schema (providing a formal event description), frequency, and a human-readable description. This aspect of the data model ensures that the system is not only structurally sound but also capable of real-time interaction and response to changing conditions within the SHM environment.

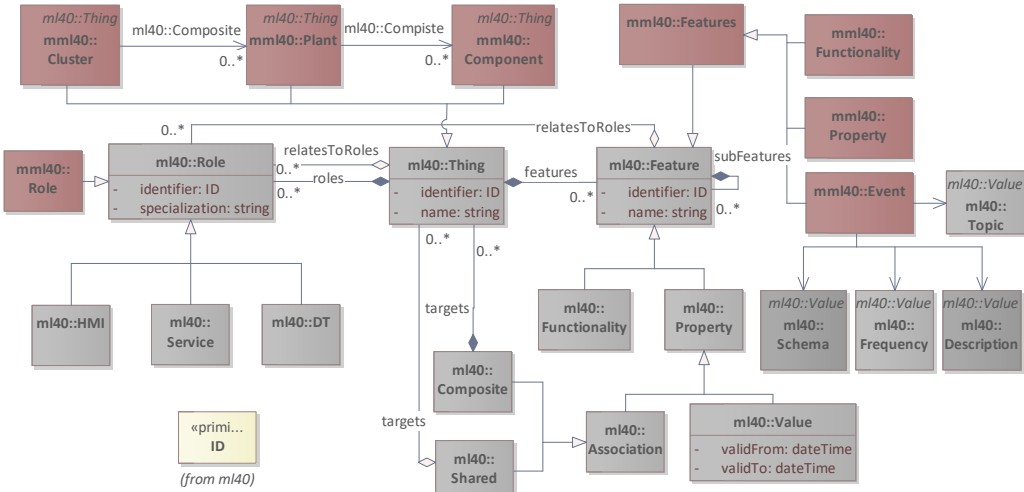

**Figure 2.** The Proposed extension of ML 4.0 [54] (gray) for the structural–mechanical domain (red).

### 3.3. Decentralized Communication and Security

In our IoT-based SHM system, despite its hierarchical structure, communication with users and service invocation is streamlined through unified interfaces and message-based protocols. This design approach ensures that technical heterogeneity within the system is effectively addressed, making communication seamless and consistent across different system levels.

Figure 3 depicts the proposed conceptual architecture of the IoT-based system. Interaction with individual objects within the system is secured through robust authentication and authorization mechanisms. Every interaction necessitates a valid authentication, authorization, and data encryption and signing process, guaranteeing the security of communications at the IoT level. To facilitate decentralized interactions within the system, we have selected the S3I as the preferred IoT infrastructure. The S3I Identity Provider offers an authentication service using OAuth 2.0. The outcome of this process is an access token representing the granted permissions, which can be used to access the S3I Services and decentralized interconnected DTs, services, and apps. An authorization system consisting of a policy engine and a policy model (e.g., Role-based Access Control) can be employed to perform the authorization against requests and the associated access tokens.

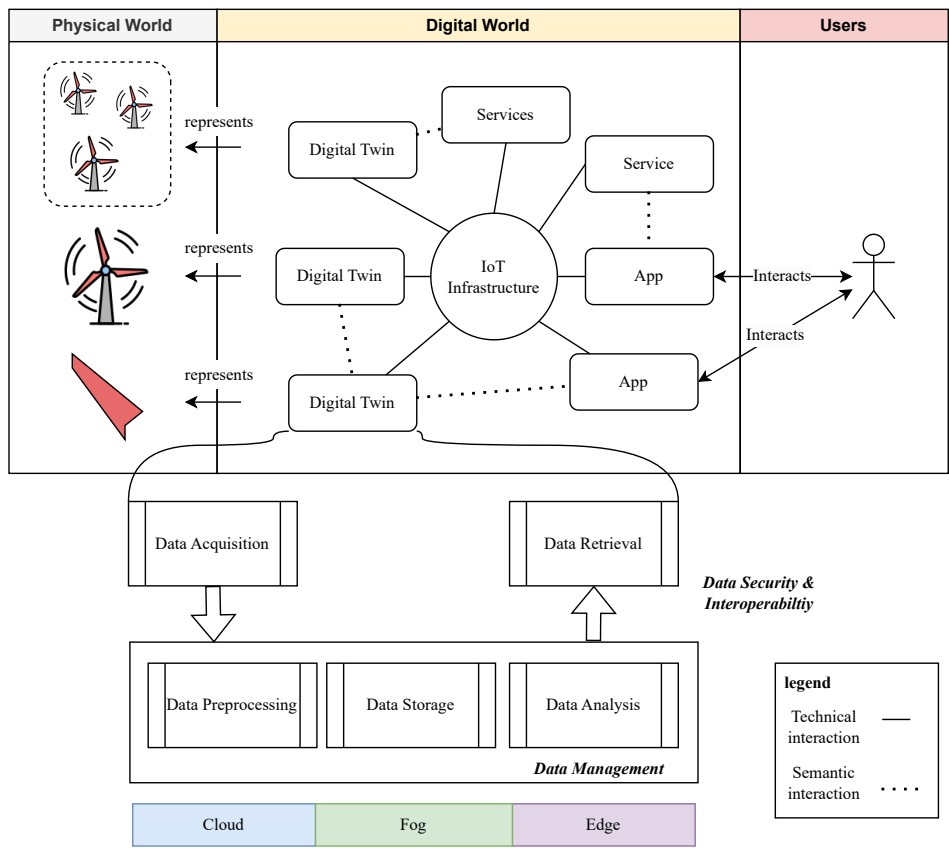

**Figure 3.** Conceptual architecture of the IoT-based SHM system with a communication infrastructure.

For communication within the SHM system, a message-based communication protocol, the S3I-Broker (S3I-B) protocol, is employed. This protocol supports data encryption and signing using RFC 4880 [60]. This means that each message is encrypted (using the private key of the sender) and signed (using the public key of the receiver) before sending. Public keys are included in the information model, which is stored in the S3I Directory. This protocol encompasses various message types, including user messages for direct communications, attribute messages for querying data, and service messages for invoking services within the system. For instance, at the component level, potential service requests might include querying for overload events, load curves over a specified period

$(F(t); \ t \in [t_1, t_2])$, sensor evaluations in specific regions ($\varepsilon_{s_1, s_2}(t, x); \ t \in [t_1, t_2]; \ x \in [x_3, x_4]$), or the number of experienced load changes. Moving beyond the component level, the plant or cluster levels can also request services, such as temperature profiles over the system's runtime or retrieval of reference or simulation data from stored databases. These outputs, along with user requests, are typically presented via a human–machine interface, such as an app.

Event-Driven Communication in the SHM System

We further extend the S3I-B protocol by an event system capable of realizing user-specified and event-driven communication. This system supports event exchange irrespective of the physical location of the things (in the cloud or on edge devices), underlining its flexibility and scalability.

The S3I-B Event system is distinguished by its thing-centric design, granting each thing the autonomy to define event content, frequency, and triggering conditions. This level of customization enables things to tailor event management to their specific needs, with the entire process efficiently managed by the DT. The protocol is thus augmented with event messages, as depicted in Figure 4. We categorize events into two types: named events and custom events. Named events are predefined by the DT and described in the data model. Custom events are requested by subscribers and emitted based on user-specified rules, such as an attribute crossing a threshold. Each event is associated with a specific topic, forming the basis of networking and indicating the event's focus.

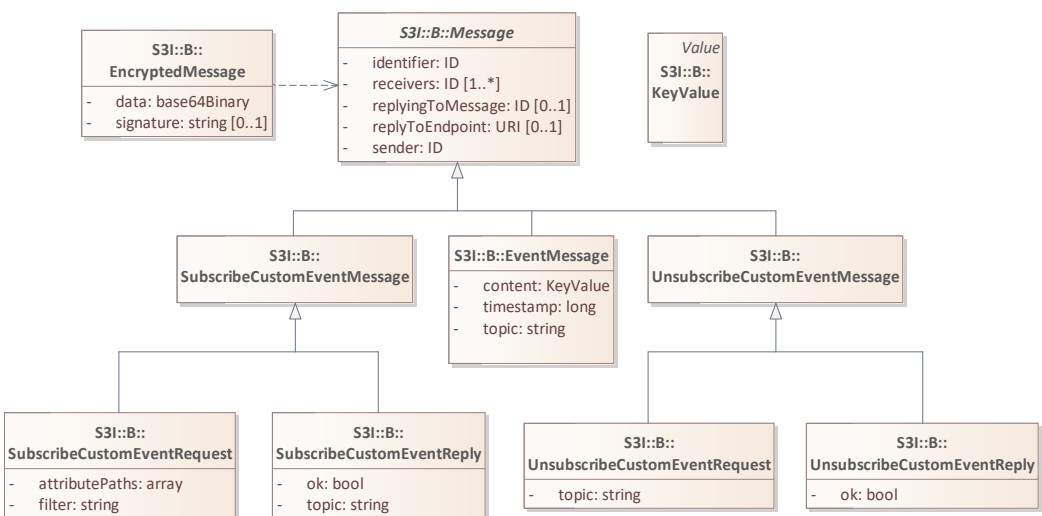

**Figure 4.** The data model for S3I-B event messages in a UML class diagram.

## 4. The Digital Cantilever in the IoT-Based SHM System

This section presents a prototypical application for a structural–mechanical cantilever beam connected with the IoT S3I infrastructure. We detail the technical implementation process, classifying the cantilever beam at the component level within our developed hierarchical infrastructure. Additionally, we provide a human–machine interface (HMI) for intuitive interaction with the monitored component. The choice of a cantilever beam as the base for our prototype is rooted in its fundamental nature. A cantilever beam is one of the primary structures frequently employed in the construction of cantilever-type components, including rotor blades, wings, and cranes. From a structural mechanics perspective, employing the cantilever beam allows us to investigate and analyze the foundational principles at its core. By isolating the beam as the smallest element, we gain a comprehensive understanding of its behavior and mechanical characteristics. Moreover, the proposed metadata model and unified communication interfaces not only facilitate this in-depth examination but also present an exciting opportunity for the expansion of these elemental structures. This expansion delivers possibilities for the development of

more intricate and sophisticated structural configurations, demonstrating the versatility and adaptability inherent in our approach.

### 4.1. Setup

The to-be-monitored component is a cantilever beam structure (see dimensions in Figure 5) that can be described based on the assumptions of the Bernoulli hypothesis [61]. A strain gauge is installed on the beam to measure the strain value. Its resistance varies with the beam's deformation. We assume that the cantilever beam is a simple and homogeneous structural–mechanical component with a low complexity. Thus, we only consider the deformation along the x-axis. Advanced sensor configurations and applications are discussed further in Section 5.

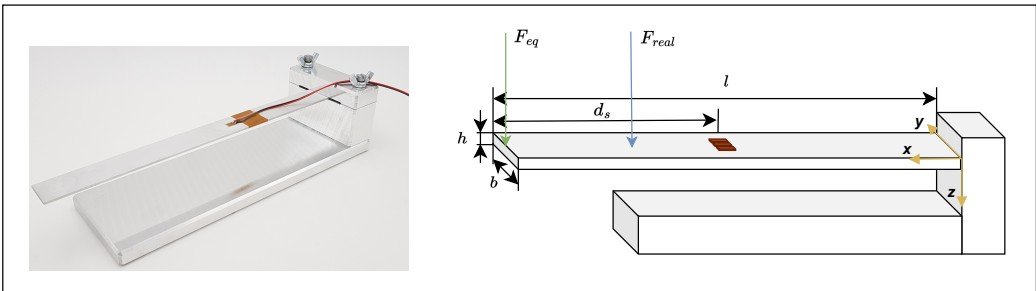

**Figure 5.** Physical Structure, the cantilever beam with one strain gauge (**left**) and its dimensions (**right**) with experimental parameters $h = 2$, $b = 25$, $d_s = 100$, $l = 220$ in mm.

### 4.2. Structural Model and Computation

The initial prototype is built with only one sensor that can only determine one unknown variable in the system. However, the load introduction point and the magnitude of the applied force $F_{real}$ are two independent effects that cannot be detected separately with one sensor. Therefore, an assumed load introduction point at the end of the beam and an equivalent force $F_{eq}$ are used for the calculation of the beam's deformation in this case.

This simplification is permissible for a phenomenological representation of the structural behavior (demonstrator) but should be replaced by an unambiguous determination of force and force application point for technical purposes, see Figure 5.

Based on the assumptions for uniaxial bending, we derive the relationship between a strain value $\varepsilon$ and the equivalent force $F_{eq}$ with Young's modulus $E$, area moment of inertia $I_y = \frac{bh^3}{12}$, cross-section geometry $b$, $h$, $l$ and sensor position $d_s$ , see Equation (1).

$$E \cdot \varepsilon(x) = \sigma(x) = \frac{M_y(x)}{I_y}\frac{h}{2} = \frac{F_{real}(l - \Delta x_F - x)}{I_y}\frac{h}{2} = \frac{F_{eq}(l - x)}{I_y}\frac{h}{2} \tag{1}$$

The strain value $\varepsilon_{ds}$ measured by the sensor corresponds to the strain at the position $x = l - d_s$. Therefore, the following applies for $\varepsilon_{ds} = \varepsilon(l - d_s)$, see Equation (2):

$$E \cdot \varepsilon(x) = \frac{F_{eq}d_s}{I_y}\frac{h}{2} \tag{2}$$

Rearranged and written for each time step $t$, it follows:

$$F_{eq}(t) = \frac{2EI_y}{hd_s}\varepsilon_{ds}(t) \tag{3}$$

Using the provided equivalent force $F_{eq}$, further information for the cantilever beam, for example, globally (approximated) deflection $w(x, t)$ (see Equation (4)) and maximum stress $\sigma_{max}$ (see Equation (5)) in the clamping, can be calculated.

$$w(x, t) = \frac{F_{eq}(t)}{6EI_y}(3lx^2 - x^3) \tag{4}$$

$$\sigma_{max}(t) = \sigma(x = 0, t) = \frac{F_{eq}(t)lh}{2I_y} \tag{5}$$

In the chosen modeling approach, we tolerate inaccuracy in the calculated deflection (see Equation (4)), which represents the deformation of a cantilever for the load introduction point at the end of the beam. In reality, however, the beam does not have a cubic but a linear deflection curve from the actual force application point. The error only occurs in the range $x > l - d_s$ and increases with increasing distance of the actual force application point from the end of the beam. The reduced complexity of the calculation approach makes it possible to capture the global behavior of the structure with little computational and sensing effort. In principle, the more detailed and local the modeling, the more complex the modeling must be, and the more computing power is required for each time step. An alternative method for real-time calculation of deformation of the cantilever beam structures is presented in [62].

### 4.3. Communication Architecture

The overall communication architecture is illustrated in Figure 6. By means of the S3I, the cantilever beam, its corresponding DT, a simulation service, and the visualization app are connected to enable situation-specific choreography. In the following, we present several hardware and software components involved in the architecture.

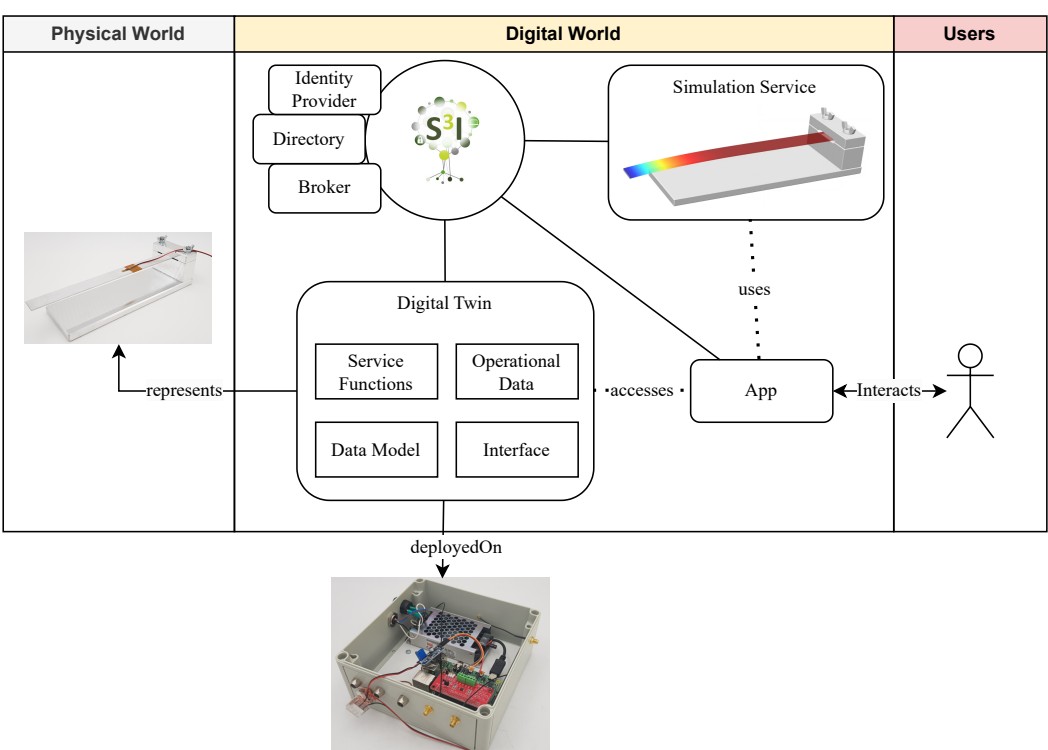

**Figure 6.** The communication architecture to monitor the cantilever beam.

### 4.3.1. Physical Twin

In the IoT, a physical twin (PT) is an entity that exists in the physical world. Here, the PT refers to the aluminum beam with one installed strain gauge, see Figure 5, measuring the strain value $\varepsilon_{ds}(t)$ at the given position $d_s$.

### 4.3.2. Digital Twin

The DT in this application represents the cantilever beam, formally described using the extended ML 4.0. The formalization is considered to be a basis for interoperability. The DT is modeled in an object diagram, as interpreted in Figure 7.

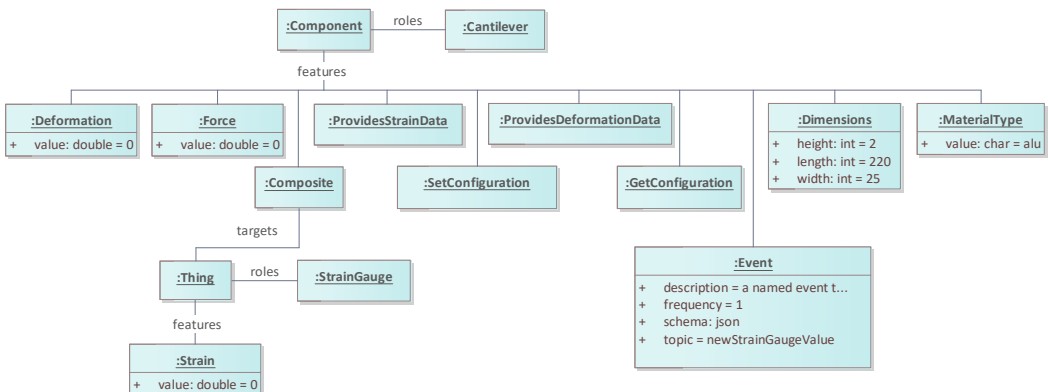

**Figure 7.** A ML 4.0-based data model of the cantilever beam.

As assigned to the role *mml40::Cantilever*, this DT is associated with various value properties such as the strain value (*mml40::Strain*) measured by the strain gauge. The property (*mml40::MaterialType*) states the type of material used to construct the cantilever beam. The beam's geometry is described by *ml40::Dimensions*.

The numerical equations introduced in Section 4.2 are the base for the implemented software services that provide strain values (*mml40::providesStrainData*) and calculated deformation (*providesDeformationData*). Both services are integrated into the beam's DT and, thus, can be called through the unified interfaces via the IoT, realizing a so-called passive DT. The composition object (*ml40::Composite*) implies a compositional relationship between the cantilever and the strain gauge that directly manages the strain value, which is represented using *mml40::Strain*.

In addition to storing and representing the current status, the developed DT also hosts a calculation service dedicated to converting the measured strain values into the equivalent force and at the beam end and the deflection, using the introduced equations in Section 4.2. The use of the event system (introduced in Section 3.3) enables the realization of an active DT. Here, DT can "recognize" each measurement of new strain value and notices all subscribers in the form of an S3I-B event message. The overall technical implementation is performed using the ML 4.0-based python framework [63], which provides the DT with a software runtime environment to ensure IoT connectivity.

### 4.3.3. Edge Device

The DT is developed as an edge DT. This means that the DT "lives" in an edge device localized near the beam. This device provides the hardware runtime environment for the DT. The edge approach empowers SHM systems to maintain functionality even in an internet connection failure. Thanks to the DT's modularization, reliable operation of sensors and data processing are separated from the Internet module, ensuring uninterrupted monitoring and analysis capabilities offline. In our application, we integrate all the needed hardware components into a compact box entitled DT Box, see Figure 6.

The corn component of the DT Box is a Raspberry Pi 4, which acts as a computing unit. It is powered by a power supply module, which consists of a 4-pin interface and a DC/DC converter. This converter ensures that the box can be supplied with a stable, suitable voltage from a variety of input voltages from 24 V to 220 V. To measure the strain value, a voltage must be applied across the strain gauge. Here, a variable current (i.e., an analog value) is generated. With an A/D converter, the current is converted to a digital value, which can subsequently be transferred directly to the Raspberry Pi for further processing via a serial communication protocol, I2C. We also installed an antenna connector in the DT Box for better Wi-Fi connectivity.

### 4.3.4. Simulation Services

Simulation services often serve as a tool for the development, validation, and verification of algorithms. In SHMs, simulations aid in predicting the future behavior of physical structures and identifying potential issues before they become critical while considering cost-effectiveness. In this prototypical application, we introduce and deploy two simulation services. The first one refers to uniaxial load estimation. In the image, a downward force is applied at the end of the cantilever beam, resulting in deformation. In the first load estimation service, we specify the x-coordinate. Based on that, the simulation service estimates the load received at that point. The second service is concerned with maintenance estimation. Here, we simplify the whole process of estimation and give only the load applied to a specific x-coordinate. From this, the service estimates the remaining useful life of the cantilever beam under these conditions. Analogous to DTs, both services are so-called passive communications participants in the IoT-based SHM system, providing interfaces for interoperable service retrieval.

### 4.3.5. IoT Infrastructure (S3I)

The use of the S3I necessitates the registration of identities. Concretely, the cantilever and the app are assigned a client ID and secret as credentials. Each user must register an S3I account to perform Single-Sign-On at the app. The result of the authentication and authorization process is an access token. The token must be provided for each interaction to ensure communication security.

In ML 4.0, there are two mappings of the data model to JSON: (1) mapping on a metadata directory (stored in the S3I Directory) and (2) mapping on a runtime environment (stored in the edge device, i.e., DT Box). The use of the provided mappings delivers an overview of the thing both at a meta-level as well as a human- and machine-understandable level. In addition, we apply the event system, which allows the DT of the cantilever beam to perform near real-time conditional monitoring. For instance, the DT of the beam generates and proactively emits event messages (see Listing 1)—that incorporate the critical information about states—with a pre-configured time frequency.

**Listing 1.** An example of an event message in json format that denotes the current operational values of the cantilever beam

```
1  {
2    "sender": "s3i:cantilever_beam:4711",
3    "identifier": "s3i:event_message:4711_a",
4    "messageType": "eventMessage",
5    "topic": "s3i:cantilever_beam:4711.newStrainGaugeValue",
6    "timestamp": 1632174117394,
7    "content": {
8      "strain": -4.161e-06,
9      "force": -0.029774266666667,
10     "deformation": [-3.462876668e-06,-1.38330135e-05, ...]
11   }
12 }
```

### 4.3.6. User and App

To provide a user-friendly interface to configure and monitor the cantilever beam, we built a Flutter-based app that also acts as a decentralized thing connected to the S3I, see Figure 8.

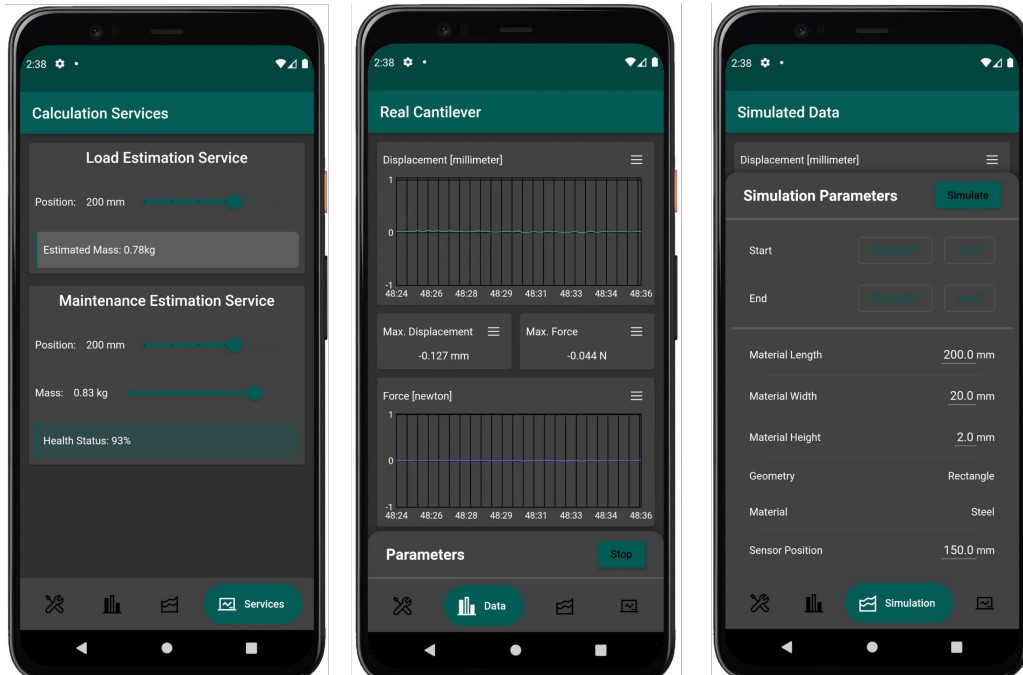

**Figure 8.** An app for the intuitive interaction with the cantilever beam.

Like the DT of the cantilever beam, the first step to use the app is authenticating and authorizing before a user needs to start data exchange with the cantilever beam. The interaction via S3I requires an S3I-compliant interface, which is directly integrated into the app. In our use case, the configuration parameters (e.g., geometry, sensor position, and material) can be transferred from the app to the cantilever using an S3I-B service request (typed as *mml40::SetConfiguration*). Accordingly, those parameters can also be retrieved with *mml40::GetConfiguration* as both are available services provided by the DT. To monitor the state change, the app subscribes to the events that are triggered by the DT of the cantilever beam if new measurements are available. After receiving these events, the app intends to visualize them in various diagrams.

## 5. Discussion

Building on the established example, different variations are conceivable to extend the functionality and applicability of the proposed IoT-based SHM system. These variations include changes in sensor configuration, the type and the number of components, and the execution platforms.

### 5.1. Variations of Sensors

An immediate enhancement involves adding additional strain gauges to the beam. For example, with an extra strain gauge along the y-axis of the beam, force and force application points can be determined simultaneously. In this context, the extension can be made from load monitoring to damage monitoring. Utilizing sensitive structural damage indicators, such as zero-strain trajectories [64], can significantly enhance monitoring capabilities, particularly for detecting cracks or other forms of structural damage. Here, a sensor setup as presented in [65] can be referenced.

### 5.2. Variations of Components

In addition to the aforementioned cantilever example, an extension of the proposed IoT-based SHM system for other structural components is possible. This adaptability allows for monitoring larger structures like a construction crane or a wind turbine rotor blade. To effectively monitor large structures using a strain-based SHM approach, several distributed measuring points become essential, as damage typically affects the strain field locally. This

requires a comprehensive network of sensors to capture these localized variations. In response to this, large structures integrate distributed sensors to form structurally and hierarchically more complex models, establishing systems at the plant layer. These systems operate as individual entities in the IoT, facilitating the centralized processing of distributed strain signals. An illustrative example is depicted in Figure 9, showcasing an emulated crane. The crane is a compositional aggregation of cantilever beams. Here, various virtual sensors are installed on the crane, delivering strain values during the operation of the crane. The crane's DT centralized processes strain values, and thus, operational data such as deformation and force at the crane's end are calculated. The access to the DT via the IoT infrastructure enables near real-time monitoring of the emulated crane with an app depicted in Figure 9. Similar ideas can be applied to retrofit more complex cantilever-typed systems, in which the basic element is always the cantilever beam. Overall, using the proposed approach not only enhances the overall availability of the health status of structures but also provides potential for efficient management of diverse components within a larger structural framework.

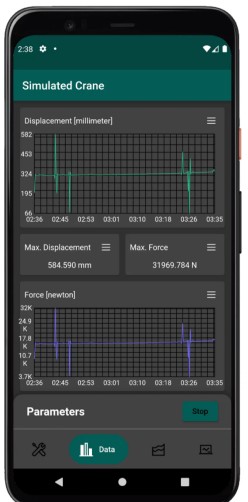 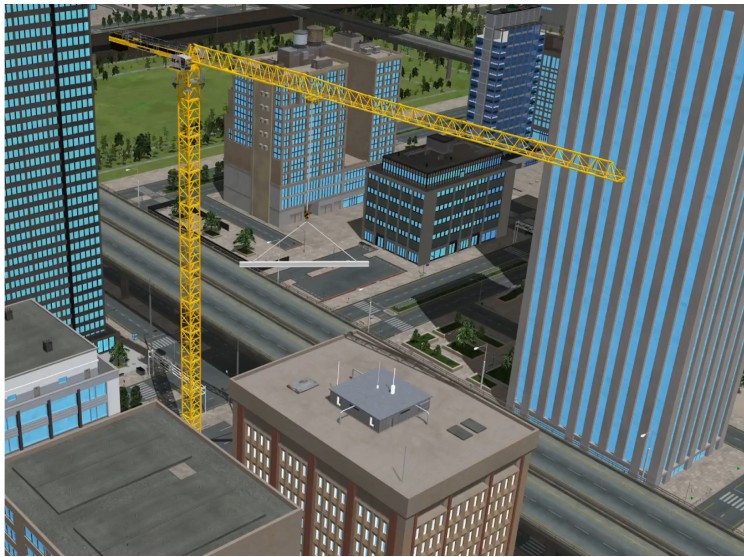

**Figure 9.** An emulated crane (**right**) built in a 3D simulation software app (**left**), visualizing the simulated crane's operational data.

### *5.3. Variations of Execution Platforms*

The implementation of DTs can be realized through different technical approaches. This flexibility allows for realizing DTs either (1) directly at an edge device that is near the assets (Edge DT), (2) to execute within a cloud service (Cloud DT), or (3) as a hybrid model in between, known as Fog DT. The diversity in execution platforms is intentionally unrestricted. Assets, which represent physical objects with limited computation and communication resources, always reside in their DTs in an edge device. Conversely, assets in the form of virtual objects, encompassing data sets and algorithms, deploy their DTs in the cloud.

In addition to the presented application in Section 4, we also have the option to relocate the DT of the cantilever beam to a cloud service. In this case, we only need to integrate the interface on the cantilever's side, ensuring the synchronization between both the physical cantilever and its cloud DT. This variation becomes particularly meaningful for applications requiring robust computing performance. Furthermore, we introduce the possibility of splitting the DT into two distinct parts. The first part resides at the edge device, which is responsible for collecting and preprocessing strain values. Meanwhile, the second part, encompassing the data model and services to calculate corresponding forces and displacements, is hosted in a cloud service. This dual-partitioning of the DT allows for a distributed approach, leveraging the strengths of both edge and cloud computing.

Overall, Fog DTs serve as solutions for applications demanding a balance between real-time processing and computational complexity.

## 6. Conclusions

This paper introduced a novel organizational scheme for IoT-based SHM systems, leveraging the capabilities of DTs. Grounded in the comprehensive literature review, our proposed framework integrates structural components into systematically organized clusters, enhancing the SHM process through IoT infrastructures.

The cornerstones of the presented infrastructure are its flexibility and extensibility, enabling the adaptation of objects with varying scopes and scales. A basic element of our approach is decentralized networking via the S3I communication infrastructure. The S3I provides all communication participants with a globally unique identity and services for authentication and authorization, ensuring secure and interoperable data exchange as well as service calls. Moreover, the existing ML 4.0 modeling language is extended to encompass special properties and service functions required for SHM applications, therefore establishing a comprehensive data model that delineates the structure and content among physical assets and their DTs. This shared formal data model increases interoperability among participants and allows the addition of semantic information to further ease the integration of new components into SHM systems.

To validate our framework, we implement a prototype using a mechanical system built with a simple cantilever beam. This demonstrator shows its capability to deliver live strain, force, and deformation data to an app in real time via its DT. This data access is only possible after users have successfully authenticated themselves and granted the appropriate authorization. The actual data exchange is mediated by the S3I Broker. This demonstration further illustrates the hierarchical concept, as the DT of the cantilever includes the information and algorithms to process raw sensor data and outputs meaningful monitoring and analysis data. Finally, the demonstrator shows the flexibility with which different software components can be deployed. In this example, the cantilever's DT is being deployed on an edge device, while simulation services are deployed on a server, and the user interface is running locally on the user's smartphone. This infrastructure can be extended to other structures or sensor configurations. With this, the functional test is completed.

However, to validate the framework's versatility and applicability in various contexts, additional case studies in various domains, such as bridges, high-rise buildings, and industrial equipment, are necessary. These case studies will help in establishing the system's adaptability to different structural complexities and environmental conditions.

In conclusion, our research indicates that SHM systems can effectively be transferred into IoT solutions using our organizational scheme.

**Author Contributions:** Conceptualization, J.C., J.R. (Jan Reitz) and R.R.; methodology, J.C, J.R. (Jan Reitz) and R.R.; software, J.C. and J.R. (Jan Reitz); validation, J.C. and J.R. (Jan Reitz); formal analysis, J.C., J.R. (Jan Reitz) and R.R.; investigation, J.C. and J.R. (Jan Reitz); resources, J.C. and R.R.; data curation, R.R.; writing—original draft preparation, J.C., J.R. (Jan Reitz) and R.R.; writing—review and editing, J.C. and J.R. (Jan Reitz); visualization, J.R. (Jan Reitz); supervision, K.-U.S. and J.R. (Jürgen Roßmann); project administration, K.-U.S. and J.R. (Jürgen Roßmann); funding acquisition, K.-U.S. and J.R. (Jürgen Roßmann) All authors have read and agreed to the published version of the manuscript.

**Funding:** This research was funded by the European Regional Development Fund (ERDF) and supported by the state of North Rhein–Westphalia under grant number EFRE-0200458 (EFRE.NRW project Kompetenzzentrum Wald und Holz 4.0).

**Institutional Review Board Statement:** Not applicable.

**Informed Consent Statement:** Not applicable.

**Data Availability Statement:** Data are contained within the article.

**Conflicts of Interest:** The authors declare no conflicts of interest.

**Abbreviations**

The following abbreviations are used in this manuscript:

| | |
|---|---|
| AD | Analog Digital |
| CIA | Confidentiality, Integrity, and Availability |
| DC | Direct Current |
| DT | Digital Twin |
| F4.0 | Forestry 4.0 |
| ForestML 4.0 | Forest Modeling Language 4.0 |
| JSON | JavaScript Object Notation |
| HMI | Human–Machine Interface |
| I$^2$C | Inter-Integrated Circuit |
| IoT | Internet of Things |
| MQTT | Message Queuing Telemetry Transport |
| PT | Physical Twin |
| REST | Representational State Transfer |
| S3I | Smart Systems Service Infrastructure |
| SHM | Structural Health Monitoring |
| UML | Unified Modeling Language |
| WSN | Wireless Sensor Network |

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
