# Peer review of "IoT-Based SHM Using Digital Twins for Interoperable and Scalable Decentralized Smart Sensing Systems"

_information, doi:10.3390/info15030121_

Round 1

Reviewer 1 Report

Comments and Suggestions for Authors

This article presents a new framework aiming at managing Wireless Sensor Networks and more globally Internet of Things for Structural Health Monitoring through Digital Twin use. If the works presented here have an interesting novelty dimension, the reviewer thinks that Digital Twin is not sufficiently introduced and need further development. Are follows the different elements that would need (according to the reviewer) a specific introduction of DT in IoT architecture for SHM :

  • line 53, DTs are presented as a main element of the proposed framework but no explanation is made to demonstrate that DT have quality to solve drawbacks developed in the previous lines.
  • In section 2.2, the state of the art that covers IoT-based SHM Systems does not introduce nor covers DT integration in IoT Systems for SHM. Consequently in section 2.3, when authors covers IoT infrastructures, the scientific reader can be lost since some solutions that are presented integrates DT while other not. Since DT has not been introduced specifically with a specific section (such as secure communication or decentralized approach for example), it is very difficult to have a comprehensive view of this section 2.3. In the same idea, DT should be a line of the table 1 since it is a key element of comparison of presented IoT infrastructures.
  • Line 253, DT is presented as the the central abstraction. If not presentation is made before, the development of the concept lacks here the fundamental understanding of what DT is.
  • The ambiguity of DT is also found in figure 2 (extension of ML 4.0) and when in figure 3 the conceptual architecture is presented. According to a major part of literature on DT, a digital twin is an interactive and continuous exchange between sensor-equipped systems and numerical model (often 3D model representations). In the approach presented here, the digital model is reduced to a simple set of numerical equation. Once again, a proper and detailed section dedicated to DT, its definition (3D model or simple set of equation),its stakes, its fundamentals and its perspective in the SHM monitoring would enhance the understanding of the reader and will help to highlight the innovative approach developed here.

  The second element of the proposal that could be improved is the demonstrator itself. Indeed, the demonstrator used is a « simple" cantilever beam that in the scope of this paper has two drawbacks : it is a very basic element compared with the example taken of the wind turbines and its blades. The second drawback is that the framework is then demonstrated through a unique element whereas IoT and WSN strength (and challenges) comes from the network and the multiplicity of the nodes. The framework presented here is supposed to help to mange such networked system but the demonstrator here does not permit to validate the hierarchical approach (only one cantilever beam, no system above) nor the network approach (and then services and IoT structure presented in the framework). The presented framework seems to have interesting perspectives but the fact that the demonstrator used is much too « simple » (from a systemic point of view) in regard of the scale of IoT architecture and the ambition presented in summary and introduction  does not permit to fully validate the approach.   Furthermore the specific elements could be improved :

  • can the authors format the reference in the following manner "author et al [XX] subdivides… » instead of « [XX] subdivides… » (line 83, line 88 etc…)
  • the interoperability is defined line 46 as "how can heterogeneous data can be processed" whereas in section 2.2.1 it is presented as the capability to connect objects to communicate (which is the common admitted definition)?
  • line 171 the authors could add some references about IoT and cybersecurity that are more globals than DoS such as for example "R. R. Krishna, A. Priyadarshini, A. V. Jha, B. Appasani, A. Srinivasulu, et N. Bizon, « State-of-the-Art Review on IoT Threats and Attacks: Taxonomy, Challenges and Solutions », Sustainability, vol. 13, no 16, p. 9463, août 2021, doi: 10.3390/su13169463. » or any equivalent.  

  To summarize, the works presented here seems very valuable but need improvement on two points

  • the fact that Digital Twin is not introduced as concept, definition and reviewed with a minimal state of the art in regard of IoT architecture for SHM
  • the simplicity of the cantilever beam in regards of IoT architecture stakes and the developed framework

It leads the reviewer to think that this submission should be improved to highlight the results of this significative work

Author Response

Dear reviewer,

thank you for reviewing our manuscript. Please find our point-to-point answers in the attached pdf file. 

Reviewer 2 Report

Comments and Suggestions for Authors

Please find below some comments to improve the manuscript:

1) At the beginning of the Introduction, the authors should be provide more concrete examples of emerging smart sensing systems and their possible applications. In its current form, the discussion is too succinct and misses to highlight the important and recent research directions in this field. For instance, environmental monitoring is a crucial field where smart sensing systems could provide a strong contribution. From a quick search on Google Scholar, some interesting and recent works on this topic can be found:

- "A Unified Bayesian Framework for Joint Estimation and Anomaly Detection in Environmental Sensor Networks", IEEE Access, 2022.

- "Advances in smart environment monitoring systems using IoT and sensors", Sensors, 2020.

2) The length of the related work section is unbalanced compared to the remaining parts of the manuscript. Please shorten it.

3) The structural models in Section 4.2 needs to be better described, possibly using more mathematical details for a more clear understanding.

4) In the Conclusion section, please provide a more thorough discussion with more concrete examples of possible future extensions of the present work.

Comments on the Quality of English Language

N/A

Author Response

(The authors gave the same response as above.)

Round 2

Reviewer 1 Report

Comments and Suggestions for Authors

The authors have partially taken into account the reviewer's comments. If the reviewer is not really convinced by the explanations on the demonstrator, he truly understands the difficulty of experimenting a more systemic testbench and considers that modifications on the paper clarifies the impact and the value of the demonstrator.

On the first improvment suggested by the reviewer (a better introduction of DT), the reviewer appreciates the addition of the dedicated section 2.2 that establish a short description of DT with some valuable and pertinent references. Nevertheless Section 2.2  is still minimalist. The reference to the global review of Jones et al is a solid reference but the authors expect that the reader refers himself to the work to understand the key fundamentals of DT. The reviewer suggests that the author should proceed to a synthetic abstact of characteristics and develop a little more this section 2.2. Other said, the authors adds references on DT but do not state their own specifications of what DT is.

Author Response

Dear Reviewer,

thank you again for taking time to review our manuscript. 

To your comment "The authors have partially taken into account the reviewer's comments. If the reviewer is not really convinced by the explanations on the demonstrator, he truly understands the difficulty of experimenting a more systemic testbench and considers that modifications on the paper clarifies the impact and the value of the demonstrator."

Thank you! 

To your comment "The reviewer suggests that the author should proceed to a synthetic abstact of characteristics and develop a little more this section 2.2. Other said, the authors adds references on DT but do not state their own specifications of what DT is.": 

Thanks for pointing this out. We fully agree with this point. As described in the manuscript, aspects and concepts to shape DTs' definition are very diverse. In our perspective, we mainly consider DTs as communication nodes in the IoT system, which use the metainformation of  DTs to realize diverse interconnections to exchange data and services. Here, we primarily focus on the data model that represents, among others, the physical structure, service functions, access interfaces, and so on. Please find our revision in Section 2.2. 

Please don't be weirded out by the fact that we left the last revision marks on the manuscript. We did so because then a brief overview of all the changes that happened in both iterations can be easily seen. 

Best regards

The authors

Reviewer 2 Report

Comments and Suggestions for Authors

The authors correctly addressed all my comments.

Comments on the Quality of English Language

N/A

Author Response

Dear Reviewer,

thank you again for all your comments and the time taken to review our manuscript!

Best regards

the authors